# Potential Endogenous Cell Sources for Retinal Regeneration in Vertebrates and Humans: Progenitor Traits and Specialization

**DOI:** 10.3390/biomedicines8070208

**Published:** 2020-07-12

**Authors:** Eleonora N. Grigoryan

**Affiliations:** Koltsov Institute of Developmental Biology, Russian Academy of Sciences, 119334 Moscow, Russia; leonore@mail.ru; Tel.: +7-(499)-1350052

**Keywords:** retina, regeneration, cell sources, differentiation depth, cell plasticity, gene expression, epigenetic features

## Abstract

Retinal diseases often cause the loss of photoreceptor cells and, consequently, impairment of vision. To date, several cell populations are known as potential endogenous retinal regeneration cell sources (RRCSs): the eye ciliary zone, the retinal pigment epithelium, the iris, and Müller glia. Factors that can activate the regenerative responses of RRCSs are currently under investigation. The present review considers accumulated data on the relationship between the progenitor properties of RRCSs and the features determining their differentiation. Specialized RRCSs (all except the ciliary zone in low vertebrates), despite their differences, appear to be partially “prepared” to exhibit their plasticity and be reprogrammed into retinal neurons due to the specific gene expression and epigenetic landscape. The “developmental” characteristics of RRCS gene expression are predefined by the pathway by which these cell populations form during eye morphogenesis; the epigenetic features responsible for chromatin organization in RRCSs are under intracellular regulation. Such genetic and epigenetic readiness is manifested in vivo in lower vertebrates and in vitro in higher ones under conditions permissive for cell phenotype transformation. Current studies on gene expression in RRCSs and changes in their epigenetic landscape help find experimental approaches to replacing dead cells through recruiting cells from endogenous resources in vertebrates and humans.

## 1. Introduction

Disorders that affect the structure and functions of the retina, usually associated with ageing or diseases, may eventually result in the partial or complete loss of vision. Currently, a large proportion of the human population faces this health problem, which is further aggravated by the increase in both life expectancy and load on the visual system. As regards the existing methods of treatment of degenerative retinal diseases associated most frequently with the death of photoreceptors (age-related macular degeneration (AMD), retinitis pigmentosa (RP), proliferative vitreoretinopathy (PVR)) and ganglion cells (glaucoma, optic nerve thrombosis) at their late stage, they have not been proven to date despite considerable effort and significant therapeutic advances. The currently known methods include drug therapy, liposomal and nanotherapy, neuroprotection, surgery, immune therapy, gene therapy, and cell transplantation, including induced pluripotent stem cells (IPSC) technology, 3D retinal organoid production, etc. The study of retinal regeneration cell sources (RRCSs), without rejecting any of the existing treatment methods, provides clues to the development and application of new approaches to retinal regeneration based on the recruitment of endogenous cell sources located in the eye tissues of animals and humans.

To date, there are no elaborate protocols to replace cells of the damaged human eye retina by recruiting certain cell populations that could be a source for its regeneration. Nevertheless, recent advances in biology that contribute to the understanding of the tissue regeneration processes, the identification of endogenous cell sources for regeneration, and also to the development of methods for inducing the differentiation of pluripotent stem cells provide a broad range of opportunities for regenerative medicine. These opportunities should also be used for the medical treatment of a retina that has been damaged by diseases or has an age-related or other pathology. The attempts to transplant foreign cells, including those obtained from induced pluripotent ones, pose some risk due to their undesirable transformations. In this regard, endogenous RRCSs are of particular interest for biomedicine.

Several criteria can be set out to include one or another eye cell population in the category of RRCSs. First, their involvement in retinal regeneration after damage or under experimental conditions has been previously confirmed. Second, experimentally revealed facts indicate the eye cells’ potency to re-express “developmental” genes and organize a permissive epigenetic landscape for this, i.e., the conditions underlying and determining the initiation and progress of regenerative responses. The RRCS criteria at the cellular level may include the ability to proliferate, migrate, and be integrated in the pre-existing retinal structure. An additional criterion is also the history of the maturation of these cell populations in development that, a priori, suggests the possibility of their reprogramming into certain cell types of retinal neurons. This does not mean that the criteria above are fully applicable to all cells considered as candidates for RRCSs. Their set depends on the species of animal, age, conditions of eye pathology or experiment, and, finally, the degree of knowledge of the issue on a particular animal model.

However, the development of biomedical approaches to provide regeneration processes through recruiting endogenous RRCSs necessitates an understanding of their biology, as well as the obtaining of information associated with this knowledge on the local and systemic factors that can support and activate the manifestation of regenerative potencies of RRCSs. A substantial number of dedicated studies in the literature consider the cellular microenvironment factors that stimulate or, vice versa, block the regenerative response of RRCSs [1,2,3,4,5,6,7,8,9,10], as well as factors exhibiting pleiotropic [11] or biphasic effects [12]. Earlier, we analyzed in detail the available information on this issue [13].

The factors that are regulators of differentiation and behavior of endogenous RRCSs, studied in vivo in various model animal objects, do not show strict universality. This is explained by the differences between the animal models selected, retinal damage conditions, methods of study, etc. However, the information obtained has significantly narrowed the search for the key external regulators of the maintenance of RRCSs’ regenerative potencies and their activation. As has been shown, these are the factors expressed during the vertebrate retina development [2,7,9,13]. In many cases, growth factors and signaling pathways, such as FGF2, EGF, IGF, CNTF, Wnt/β-catenin, Notch–Delta, etc., do not only support the special phenotypic status of RRCSs, but also activate the proliferation and reprogramming of cells in the neural direction to replace dead mature retinal neurons. TGF-β, BMP4, RA, SHH, and other signaling pathways, on the contrary, often block such cell responses. Systemic factors associated with both age and retinal damage, such as hormones, factors, and cells of the immune system and blood, have an inextricable relationship with local factors and are also regulators of the “regenerative” behavior of RRCSs [13]. Besides, there is a need to understand the state of endogenous RRCSs proper as regards the depth of their differentiation and specialization in animals from different classes and humans. 

As can be seen from various examples of vertebrate models, endogenous RRCSs can be represented by stem and low-differentiated cells, as well as by latent, differentiated progenitors. To date, several cell populations have been collectively described as candidates to be RRCSs [9,14,15,16,17,18]. These cell populations, recorded from different animal models, are schematically represented in Figure 1 and discussed below in the respective sections. Figure 2 shows schematically the main structure of the retina in vertebrates and cells that are most common damage targets.

Various categories of RRCSs were discovered not only morphologically, but also as a result of studying the gene expression profiles and using the marker proteins expressed by these cells. The molecular and genetic profiles of RRCSs reflect the characteristics (although to varying degrees) of eye primordium and retinal precursor cells. As is known, RRCSs are characterized by the expression of the transcription factors (TFs) typical of cells of the “eye field”, which is the eye formation area in the neural ectoderm of the anterior part of the neural plate in the vertebrate ontogeny. These TFs, attributed to such families of factors as Pax6, Chx, Rx, Six, Sox, Prox, Pitx, etc., are evolutionarily conservative and are referred to as eye field transcription factors (EFTFs) [19,20,21]. In addition to EFTFs, the expression of TFs used for pluripotency induction (Oct3/4, Sox2, Klf4, and c-myc) was reported for some RRCS populations [22,23,24]. Along with undifferentiated RRCSs in the circumferential ciliary zone of the eye in lower vertebrates, the latent but differentiated RRCSs, including the retinal pigment epithelium (RPE), the iris, the ciliary body (CB), and Müller glial (MG) cells, also show the expression of genes that provide their clear specialization and functioning in the retina (see below). As regards RRCSs that have, as we shall see below, a special “young” phenotype and are targets for factors triggering regenerative responses, it is necessary to take into account the age and species of animals, which also become apparent in the epigenetic features of RRCSs [25,26,27].

This review is an attempt to show that RRCSs exhibit genetic and epigenetic potencies: proliferation, reprogramming, and subsequent differentiation towards the phenotypes of affected or dead cells for their replacement. The key to the implementation of these potencies and the identification of ways of RRCS-based treatment in case of disease is the complete knowledge of both the biology of these cells and the factors that regulate cell behavior or, in other words, the ways of the induction and targeted regulation of regenerative responses. Special attention in the review is paid to the currently available relevant information on the extent to which RRCSs are advanced towards differentiation in adult vertebrates of different classes and humans and to what extent they have retained the progenitor properties in terms of gene expression and epigenetics.

## 2. Eye Ciliary Zone

### 2.1. Circumpherential Ciliary Marginal Zone

RRCSs of this category are located on the extreme periphery of the retina, in the so-called “corner” of the eye (Figure 1). The ciliary zone cells constitute a heterogeneous population, i.e., they show the expression of stemness and early progenitor features to different degrees. This region in vertebrates also varies in the size of the cell population, which is reduced in the series from fish to mammals.

In adult lower vertebrates (fish and amphibians), the cells of this region do not have signs of specialization, and their poorly differentiated status has long been known [28,29,30]. In fish and amphibians, the ciliary marginal zone (CMZ) of the retina provides the contribution of cells to the growing retina throughout the life of these animals and is represented by stem and poorly differentiated cells [31,32]. In the species of adult fish and amphibians under study, as well as in bird embryos, this region is a locality of persistent neurogenesis, which ensures the growth of the retina from the periphery during development and regeneration [29,30,31,33,34] (Figure 3). The major part of the available data on the CMZ has been collected as a result of long-term studies on models of retinal development and regeneration in some species of fish [2,35,36] and amphibians [31,37]. In these animals, the CMZ contains peripherally located stem cells and more centrally located progenitors. The former are supposed to be capable of permanent independent divisions, and the latter of only a limited number of divisions [2,38,39,40].

In the past decade, CMZ cells of lower vertebrates have been intensively studied as regards their genetic profile. In particular, the expression of TFs associated with neurogenesis was investigated. At the same time, studies considered the regulatory signaling pathways that trigger and block neurogenesis in the CMZ [2,41,42,43]. It was found that the TFs whose expression is characteristic of cells in this region and causes the manifestation of neurogenic potential belong to the group of EFTFs. Their spectrum is encoded by the *Pax6* and *Rx* homeobox genes, members of the *sine oculis* homeobox gene family, *Six3*, and by the LIM homeobox gene *Lhx*. This list can be extended with *Chx10* (*vsx2* in fish), *Crx*, and *Pitx*, which also work at the early stages of eye development [44,45]. As was reported, fish have approximately five rows of cells on the extreme periphery of the CMZ that express the *Rx* homeobox gene [46].

Tailless amphibians, such as, in particular, the African clawed frog *Xenopus laevis*, have also long been a subject of studies on the molecular and genetic properties of CMZ cells [31,37]. It has been shown that the expression of genes associated with early stages of eye development is typical for the most peripheral part of the frog eye CMZ, whereas the genes that begin to function at more advanced stages of retinogenesis are expressed in cells occupying a position closer to the center of the CMZ (Figure 3). It has also been found that in *X. laevis* tadpoles, prior to metamorphosis, CMZ-associated cells are involved in retinal regeneration after partial retinal resection [47]. This is due to the reproduction of progenitors, the cells that exhibit the expression of *Rx* and other marker genes, in the wounded area. Knockdown of the *Rx* gene disrupts the ability to regenerate the retina, which confirms the idea that the regenerative responses of RRCSs require the recapitulation of developmental genetic events.

In a recent study [48], it has been found that some genes in the CMZ of the differentiated retina in *Xenopus* tadpoles are expressed downstream of *Rax*, the eye field gene whose expression is induced during eye development by the IGF signaling pathway. The expression of *Rax* downstream genes clearly demonstrates the progenitor properties of CMZ cells in the *Xenopus* retina (Pan et al., 2018). However, little is known about the role of CMZ in regeneration in postmetamorphic tailless amphibians. In the study of Mitashov and Maliovanova [49] on adult *X. laevis*, CMZ cells were able to partially repair the retina after its removal. Retinal regeneration by CMZ cells was also observed in *X. tropicalis* [50]. 

A study of gene expression in CMZ cells in caudate amphibians, e.g., the newt *Pleurodeles waltl*, demonstrated the progenitor status of cells in this zone that are involved, along with RPE (see below), in retinal regeneration. The expression of TFs belonging to the EFTF group (Pax6, Otx2, and Six3, as well as c-myc, Prox1, and Pitx2) was also detected [51,52,53]. When we saturated adult newt CMZ cells with BrdU, a proliferation marker, this showed that 15–20% of CMZ cells were in the phase of DNA synthesis within a month after the artificial retinal detachment [54].

The contribution of the marginal zone of the retina to its growth and regeneration, as well as the size of this cell population, decreases in an evolutionary series. In reptiles, birds, and mammals, this region is significantly reduced; nevertheless, its cells retain signs of low differentiation and the ability to proliferate [34,55,56,57]. A recently published study reports that reptiles have a region similar to that of the CMZ in fish and amphibians [58]. In a comparative study of the zone homologous to CMZ (the region between the differentiated retina and the CB) in lizards and snakes, it was found that cells of this region show a proliferative potential and are a source of an insignificant eye growth. These cells express proliferation markers and the conservative genes of eye and retinal development. The same as for the retinal progenitor cells of the developing retina in other vertebrates, they express markers *Pax6*, *Vsx2*, *Rx1*, and *Hes1* (at the mRNA level), and also Sox9 and Sox2 (at the protein level) [58]. Proliferating PCNA-positive cells were previously found in the marginal region of the adult turtle retina [57].

The CMZ region of the chick eye is similar to that in fish and amphibians, but is formed before hatching and, therefore, its contribution to retinal growth is insignificant [59,60]. In post-hatch birds, despite the fact that this retinal zone does not completely disappear, the production of new neurons and glial cells in it is extremely limited [61,62,63]. However, in chicks at late stages of development, slow-maturing proneural cells that express early markers of retinal differentiation, such as HuD, calretinin, and visinin, can be found on the periphery of the retina outside the CMZ [55].

It has long been believed that mammals and humans lack the CMZ or its comparables. However, a detailed study of the retinal marginal zone cells in the embryonic development of mice has revealed limited neurogenesis with the formation of ganglion cells [64]. During development, cells on the margin of the mammalian and human retina express Pax6, Chx10, and Lhx2, as well as the Otx1, Prox1, and Pitx1-2 transcription factors, retinol dehydrogenase Rdh10, etc. [21,65,66,67]. Until birth, the retinal marginal zone cells in mice are positive for BMP4 and the cyclin D2 proliferation factor, as well as for the Msx1 and Zic1/2 transcription factors [64,67,68].

The retinal margin, represented by the pars plana (orbicularis ciliaris), a flat sheet of cells connecting the ciliary body and the retina, may retain neurogenic potential, as evidenced by experiments on the adult mice of several lines [56]. The cells formed from the single-layer epithelium in pars plana are non-pigmented and morphologically different from the CB cells. It may be referred to as either the marginal region of the neural retina or the CB. The pars plana can be regarded as the CMZ (in its accepted understanding) very conditionally. It has been found that this region of the adult mouse retina contains cells that express the *Atoh7* gene, a marker of neurogenesis. The expression of the *Atoh7* gene is known to occur immediately before and during the M phase of the cell cycle [69]. The authors [56] suggest that *Atoh7* expression is a non-cell autonomous response of the pars plana to the loss of retinal ganglion cells (RGCs) during retinal development. In response to specific the depletion of part of the RGCs, the neurogenic potential of progenitor cells in the pars plana is activated, leading to the generation of the ganglion cell population [56]. Thus, as has been found to date, the TFs whose expression is exhibited by the eye field cells during development belong to those TFs expressed in the cells of the ciliary, marginal zone of the vertebrate retina and are responsible for the manifestation of the neurogenic potential. These cells also have a proliferative potential, with its level, as well as the size of this cell population, decreasing in an evolutionary series. 

The process of retinal development and the differentiation of retinal cell types is provided by the mechanisms of interaction between intracellular TFs and extracellular signaling molecules, and also by the epigenetic regulatory mechanisms, including variations in histone modifications and the chromatin landscape. The epigenetic modifications, occurring along and in close connection with the genetic ones, are now of increasing interest to researchers who intend to study the status of RRCSs during retinal development and regeneration [26,27].

Recently, information has been published on the epigenetic status of cells in the zebrafish eye CMZ [70]. The study considers the DNA methylation in CMZ cells using Dnmt1, one of the DNA methyltransferase (Dnmt) enzymes that catalyzes the DNA methylation process [70]. It was previously known that the loss of this enzyme results in genomic hypomethylation, and that Dnmt1 is capable of maintaining the function of progenitor cells in self-renewing somatic tissues [71]. Angileri and Gross [70], using a *dnmt1* mutant zebrafish with abnormalities similar to those documented in *Dnmt1^-/-^* conditional knockout mice, have established that the function of this gene is required within the CMZ cells to exhibit progenitor properties and maintain the retinal stem cell (RSC) population. Within the *dnmt1*-deficient CMZ, a decrease in the number of RSCs, a reduced level of their proliferation, and aberrant gene expression, were detected. It can be assumed that the *dnmt1*-related epigenetic modification may be responsible for the maintenance of stem and progenitor properties in the retinal CMZ cell population in lower vertebrates. This is probably only one out of the large and still poorly understood variety of epigenetic mechanisms responsible for the progenitor status of CMZ cells and neurogenesis in retinal development and regeneration in lower vertebrates.

### 2.2. Ciliary Body

In contrast to lower vertebrates, adult mammals lack a region similar to the CMZ, capable of neuron production throughout the life of the animal, and, therefore, the regenerative potencies of the mammalian “corner” of the eye cells in retinal regeneration are extremely limited [72]. 

In adult mammals and humans, the region homologous in localization to the CMZ is represented by the ciliary body (CB) (Figure 4). The anatomical homology of the CMZ in lower vertebrates and the CB in higher ones suggests that CB cells retain some growth and regenerative capabilities. The mammalian eye CB is represented by two rows of cells: the outer pigmented layer, consisting of cuboidal cells and which is formally an extension of the RPE, and the inner layer, consisting of non-pigmented cells that directly interact with the vitreous fluid of the eye. In terms of cell specialization, the CB in higher vertebrates is a structure responsible for the production of vitreous fluid components and is involved in visual accommodation through changes in the shape of the lens, with the iris supported by the CB [73,74,75]. A mass spectrometry analysis of the proteomics profile of human CB tissue, which aimed to study the specialization and function of the CB, has revealed a large set of proteins that are specific to the CB and, to some extent, characterize its function [76]. 

In the adult mammalian retina, the CB area with possible proliferation is represented by the non-pigmented (inner) layer of cells which, however, do not exhibit progenitor features in vivo [77]. Nevertheless, in the case of damage with the loss of ganglion and amacrine cells, proliferation is activated in the non-pigmented layer of CB epithelial cells [16,72,78,79,80].

After isolation and under permissive in vitro conditions, not only adult mammalian CB cells can be activated to enter the S-phase, but their potential for neurogenesis is also manifested. The evidence of the presence of “stem” cells in the mammalian and human CB was reported in the early 2000s [81,82,83]. The epithelial cells in the pigmented layer of the mammalian CB in vitro were found to form spheres that resembled the neurospheres formed by neural stem cells (Figure 4).

The CB cells in spheres proliferated and expressed nestin and Pax6, a key TF of eye development. However, besides these markers, the marker of epithelial cells, Claudin-1, was also detected. Thus, the spheres obtained in vitro contained cells of a mixed (epithelial and neuroprogenitor) phenotype [81,82,84]. Subsequently, the cultivation of human eye CB cells showed that pigmented cells are more efficient than non-pigmented ones in terms of their ability to form neurospheres [85]. In studies on the adult mammalian CB in vitro, besides TF Pax6, protein markers of retinal neurons, in particular of photoreceptors, were sometimes detected as expressed by cells in spheres. However, the extremely low number of pigmented epithelial cells with “stemness” features in the CB, as well as their even lower production of retinal progenitors, destroyed hopes for the practical use of the CB [86]. Nevertheless, the attempts to increase the production of neurons by CB cells and the search for appropriate in vitro conditions still continue for possible practical use in the future [87,88]. 

Clarifications on the CB cells’ phenotype have shown that CB cells in vitro exhibit only some stemness features, but are not truly stem cells [89]. Cicero and co-authors [89] found that the in vitro ciliary epithelium-derived spheres, despite proliferation, do not lose the properties of differentiated, pigmented epithelial cells. They retain the pigment, the membrane interdigitation, and the epithelial cell–cell junctions. When exposed to growth factors in vitro, these cells expressed nestin and the pan-neuronal marker βIII-tubulin. Although the cells aberrantly expressed neuronal markers, they failed to differentiate into retinal neurons in vitro or in vivo. As a follow-up to these studies, an analysis of the proliferative and differentiating potencies of the CMZ in lower vertebrates and the CB in higher vertebrates has been done in a recent work [90]. The authors emphasize that, despite the fact that mammalian CB cells are phenotypically plastic to a certain extent, the question of their ability to generate retinal neurons is still open and requires further research on the mouse and human CB in vitro, as well as the assessment of the progenitor potential exhibited by CB cells in vivo [90]. 

In a study of the potencies and biology of CB cells, efforts were made to determine the epigenetic regulators of cell differentiation into retinal neurons. Using human cadaveric eye material, Justy and Krishnakumar [91] analyzed DNA methylation and histone methylation by repressing and activating epigenetic regulators in the same chromatin region (H3K4me3 and H3K27me3 marks) in the lineage commitment of retinal progenitors derived from the CB. It was found that the promoters of a number of genes responsible for the pluripotent state were methylated in a population of differentiated cells. Furthermore, bivalent modifications involved in the process of the differentiation of a CB-isolated cell line into neural and glial cells were also identified [91]. 

Thus, after the consideration of the molecular, genetic, and epigenetic properties of cells of the CMZ in lower vertebrates and the CB in the mammalian eye, with the latter being homologous to the former in localization, as actual (for the CMZ) and potential (for the CB) RRCSs, the following conclusion may be drawn. In the evolutionary series of vertebrates, not only the number of cells in this eye region is reduced, but also the level of differentiation increases from stem and low-differentiated cells in the fish and amphibian CMZ to specialized cells in the mammalian CB. The latter retain some features of their progenitors, which are, nevertheless, manifested only in permissive, specially selected in vitro conditions. These features are expressed, along with the retention of some initial traits of the epithelial phenotype, by the CB cells. 

These plastic (to a varying degree) features of the phenotypes of the eye ciliary zone cells in different vertebrates and their internal, as well as external, epigenetic regulation by the microenvironment determine the different regenerative potencies of this RRCS category in animals from different classes. It should also be noted that the ciliary region is differentiated late in the eye development relative to the equatorial and central retina, which determines its properties as a region of progenitor cells (in lower vertebrates) or cells exhibiting some progenitor features in vitro (in higher vertebrates). The spatio-temporal molecular mechanisms responsible for such regional differences in the vertebrate eye ciliary zone are discussed [90].

## 3. Retinal Pigment Epithelium

Retinal pigment epithelium (RPE) is a monolayer of hexagonal, polarized, and highly specialized cells. The RPE is located in the posterior wall of the eye between the neural retina and Bruch’s membrane, underlain by the vascular layer (Figure 1; Figure 2) [92,93]. In addition to the transfer of substances along the basal/apical gradient of cells, RPE performs a number of important functions required to maintain the homeostasis and activities of the retina. The main RPE function is to provide the process of light perception and signal transmission [92,93]. Along with the obvious specialization, RPE cells in lower and higher vertebrates and humans retain the ability to be reprogrammed into retinal progenitor cells and retinal neurons. 

For the mammalian and human RPE, these transformations are possible only under in vitro conditions, where morphological changes in cells and the reprogramming process occur upon adequate stimulation (Figure 5) [24,84,94,95,96,97,98,99,100,101]. Some authors [18,102] regard in vitro RPE cells as retinal stem cells (RPE-derived stem cells, RPESCs). After various manipulations in vitro, it was found that human RPE cells, both freshly isolated and immortalized lines, can be committed to the neural pathway of development, which is preceded by the loss of some initial phenotypic and molecular features and the acquisition of proneural ones [97,98,99,100]. When exposed to permissive in vitro conditions, adult human RPE cells reduce the expression of the specific protein RPE65 and up-regulate the expression of progenitor markers, which are the TFs Oct4 (POU5F1), Nanog, Prox1, Musashi1, and Pax6. This is followed by the expression of proteins that are markers of neuronal differentiation: tyrosine hydroxylase (TH) and neurofilaments (NFs) [98]. Salero and co-authors [101] reported that human RPE cells under permissive conditions of culture can exhibit pluripotency and proliferate, and, in addition to neural progeny, can also produce osteo-, chondro-, and adipo-lineages of mesenchymal progeny. It is also reported that neuronal spheres composed of mouse RPE cells in vitro contain reprogrammed cells that can develop in two directions: either return to the original phenotype or acquire photoreceptor differentiation [103]. The same phenotypic changes of cells obtained from RPE-derived spheroids occurred in vivo when they were transplanted subretinally into eyes with simulated retinal degeneration. In the latter case, the cells could become integrated in the structure of the retina and replace its lost cells. The Hippo signaling pathway has been found to be involved in the neural reprogramming of RPE in mice [103]. 

Continuing studies on chicken embryonic (E6) eye tissues in conditions of cell reaggregation in vitro proved to be very useful with regard to the role of RPE both in retinal development and regeneration [104,105,106]. These findings defined an inductive and decisive role of RPE for the genesis and regeneration of the retina. It was found also that the RPE effect is not independent but coupled with that of MGs and that RPE cells can be a source of the in vitro production of retinal neurons. In retina reconstruction experiments on cell reaggregates from neonatal gerbil retinas, it was established that RPE is a producer of factors promoting the formation of almost complete retinal spheres [107]. It is predicted that these results will become biomedically relevant and stem cell biology will soon open ways to produce large amounts of human retinal precursors [105].

However, the ability of native RPE cells to be reprogrammed is not observed in higher vertebrates and humans in vivo. In mammals and humans, the dissociation and exit of RPE cells from the layer is often accompanied by the epithelial–mesenchymal transition of the RPE cells involved, along with glial cells, in the development of such pathology as proliferative vitreoretinopathy (PVR), which is frequently caused by retinal rupture and detachment [108,109,110,111,112]. In this regard, for the treatment of degenerative retinal diseases in humans, in particular age-related macular degeneration (AMD), it is currently recommended to use RPE cells obtained from pluripotent stem cells rather than endogenous ones [112]. 

RPE cells of both lower and higher vertebrates exhibit all the signs of specialization, including morphological, molecular, and functional ones. Human eye RPE cells express proteins that are markers of phagocytosis, basal and apical cell surfaces, and the visual cycle, including RPE65, LRAT, and CRALBP [112]. A proteomic analysis of the human RPE has also revealed a wide range of proteins that characterize the deep specialization of this tissue, in particular the content of proteins involved in the visual cycle, melanogenesis, autophagy, etc. [113,114].

Among the studied animal models, the in vivo ability of RPE cells to be reprogrammed into retinal cells is most perfectly exhibited in caudate amphibians, such as newts (Figure 6 and Figure 7). After damage and even the removal of the native retina, the adult newt RPE becomes a source for the production of a new, structurally and functionally complete retina [115,116,117,118,119]. The main events on the pathway of this transformation are as follows: the exit of cells from the RPE layer, their proliferation with the loss of the initial phenotypic features, the formation of a population of cells with neuroblast features, and then their differentiation with the recovery to a fully functioning retina (Figure 6 and Figure 7). All these stages are described from the aspect of cell biology and variations in the expression of genes of newt eye RPE cells [51,52,53,119,120,121,122,123].

We analyzed the cellular properties and molecular genetic profile of the RPE in newts of different species and have come to the conclusion that, at the initial stage, these cells are morphologically and functionally fully specialized. However, taking into account the expression profile of a number of genes and proteins, newt RPE cells can be classified as cells with a “young” phenotype. This means that they maintain the expression of early eye development TFs (Pax6, Pitx1, and Hes1) characteristic of retinal progenitors, along with the expression of genes specific for RPE differentiation (*RPE65*, *Otx2*, and *Mitf*) [124]. The adult newt RPE exhibits the expression of *Ns*-encoded nucleostemin, a transcriptional activity regulator typical of stem and low-differentiated cells [125,126]. Recently, using the technique of *Pax6* knockdown in newt larvae, this master gene has been shown to play a significant role in the selection of the pathway: in the neural direction that occurs in newts, and in the mesenchymal direction found in the human RPE pathology in vivo and in vitro [127].

The potencies of RPE cells as RRCSs were also studied in tailless amphibians [3,128]. It was shown that, during reprogramming, the RPE cells leave the layer, migrate towards the retinal vascular membrane, express the TF Pax6, proliferate, and form a neuroepithelium which, while proliferating and differentiating, forms a new retina (Figure 6). 

In chicks, the retinal regeneration from RPE cells also occurs in vivo, but within a short timescale of development (E3.5–4.5), and the forming retina has reversed polarity [129]. Later on, the expression of the TFs Pax6 and Chx10, regulated by the FGF2-FGFR/MEK/Erk signaling cascade, was detected in the neuroblast layer formed from chick RPE progeny in the presence of FGF2 [130]. Using the chick retina regeneration in vivo model, the pluripotency factors (Sox2, c-myc, and Klf4), along with Pax6, were found in the RPE cells that had not yet begun to proliferate. However, as it turned out, the RNA-binding protein Lin-28 (as a target of FGF2) and β-catenin inactivation are additionally required for RPE reprogramming [23,131].

In a study of RPE potencies, no involvement of RPE cells in retinal regeneration could be found in fish [132]. The fish retina regenerates from CMZ cells [2,36,133]. However the ability of fish RPE cells to replace losses of their layer has recently been reported [134]. Until this study, adult vertebrate RPE cells were known to be capable of only limited proliferation on the periphery of the layer [131,132]. Recently, a work based on a transgenic zebrafish model (*rpe65a*: nfsB-eGFP) has shown damage to significant areas of the differentiated RPE layer, manifested as cell degeneration, and also to Bruch’s membrane and photoreceptors adjacent to the damaged areas of the RPE [134]. The RPE layer regenerated, with the complete recovery of its function, within one month from the periphery to the center due to the cells retained on the periphery. The Wnt signaling pathway is supposed to play a certain role in stimulating the proliferation of RPE cells. An assumption has been made that the source of proliferating RPE cells on the periphery may be the CMZ stem cells that express the TF Rx2 and are capable of differentiating in both directions: neural retinal cells and the RPE [134]. 

Thus, the obtained data suggest that the acquisition of the ability to be reprogrammed into retinal neurons by RPE cells in vivo requires the conditions of genome operation that would allow the expression of the genes and protein TFs associated with the early eye and retina development and, simultaneously, the inhibition of the expression of the genes and proteins responsible for the RPE specialization of cells [128]. At the same time, the stages of a special state—the combined manifestation of the initial properties of RPE cells and the progenitor features—are also possible in the RPE genetic profile [135,136]. The mechanisms that regulate these processes are, however, not fully understood, but the recently obtained data can shed some light on this issue.

As is known, the source of the light-sensitive neural retina and the light-absorbing RPE during development are the eye anlage cells: the optic vesicle and, subsequently, the optic cup, i.e., retinal progenitors [137,138,139]. It is also known that, in the absence of a signal from the ectoderm covering the eye optic cup, the cells of the future neural retina develop into the RPE [140]. Based on these data, Dvoriantchikova and co-authors [141] have suggested that the RPE cells may be epigenetically similar to the progenitor cells of the anlage posterior wall (optic vesicle progenitors) and, therefore, a priori, have a potential to regenerate the neural retina. In this work [141], an attempt was made to determine the epigenetic features of the adult mouse eye RPE cells associated with the ability to be reprogrammed into neurons, as well as identify the epigenetic mechanisms that block this ability in mammals in vivo. The obtained data show that, epigenetically, adult mouse RPE cells are progenitor-like. The use of the methods of microarray, ChIP-seq, and whole-genome bisulfite sequencing has provided evidence that most of the key genes associated with the progenitor phenotype are located in unmethylated chromatin, which is known to be permissive for gene expression. The state of methylomes in the RPE cells has significant similarities between embryonic and adult mice: most of the gene promoters in the mouse RPE cells were in open (active) chromatin, which is characteristic of epigenetically mobile stem and progenitor cells. What, then, blocked the ability of mouse RPE cells to be reprogrammed into retinal cells? An analysis of the methylation of the RPE genes responsible for the specification of retinal phenotypes showed the following: the genes controlling the non-photoreceptor differentiation of neurons had promoters in repressed chromatin sites located in their unmethylated or weakly methylated regions. Thus, many of the key genes responsible for the manifestation of differentiation of amacrine cell, horizontal cell, RGC, and bipolar cell phenotypes proved to be repressed. According to the authors [141], these genes can only be activated in the presence of certain pioneer TFs [142] that can initiate transcription events in the “closed” chromatin of adult mouse RPE cells. Furthermore, most genes required for differentiation into photoreceptor phenotypes (rods, cones) and the development of their function, light perception, proved to be highly methylated, as compared to the degree of methylation (low or lack of) of the genes associated with other non-photoreceptor cell phenotypes of the retina. The demethylation of the photoreceptor genes’ regulatory elements is the second mechanism required for the full manifestation of the progenitor potential and plasticity of RPE cells. The authors also suggest that both mechanisms are implemented to initiate and successfully regenerate the retina after damage in amphibians, but do not work in mammalian RPE cells [141].

In concluding this section, it should be noted that the vertebrate and human eye RPE, despite its complete morphological and functional specialization based on the RPE-specific profile of gene expression, exhibits features of phenotypic plasticity and reprogramming. The phenomenon consists of the retention of the progenitor properties detected at the level of expression of the respective genes in adult amphibians and bird embryos, which allows retinal regeneration in these animals in vivo. In adult mammals, the progenitor features of RPE cells are detected at the level of epigenetic status, which has the signs of progenitor status, and the phenotypic plasticity of RPE cells is manifested only when exposed to conditions of in vitro stimulation. The relationship between the exogenous factors that can stimulate RPE reprogramming in vitro and the changes that occur in the epigenetic features of RPE cells in this case are not fully understood, although this relationship seems to be a basis for finding ways to stimulate the regenerative responses of the RPE as an RRCS in higher vertebrates and humans in vivo.

## 4. Iris

The iris pigment epithelium has a common origin with the RPE and, in many respects, shows similarities to this tissue: it has apical–basal polarity, the presence of pigment in the cytoplasm, and tight junctions between cells and microvilli (Figure 1). Due to the results of in vitro experiments, pigmented epithelial cells of the mammalian iris are also commonly considered as an RRCS. When cultured, iris cells are known to demonstrate some features of stem cells and early progenitors [84,139]. It was found that chicken iris pigment epithelial cells isolated on day two post hatching, cultured under permissive in vitro conditions, could form neurospheres, where cells proliferated and expressed proteins that are markers of retinal progenitors [143]. After placing such neurospheres on laminin in vitro, their cells expressed neural, retinal-specific marker proteins. It is worth noting that, when neurosphere cells were co-cultured with retinal progenitor cells, the iris-derived neurosphere cells could transform into photoreceptors and Müller glial cells. These data indicate the neurogenic potential of iris cells and their ability to reprogram their phenotype [143]. Studies on the same model, chick eye iris tissue cells in vitro, have shown a significant role of the Wnt signaling pathway in the negative regulation of the reprogramming process in the neuronal direction which, in turn, proved to be dependent on the cell environment (in this case, the culture substrate). The authors suggest that the iris tissue cells have certain stem properties and use the term “stemness nature” [144]. 

The isolation and cultivation of adult pig iris cells have also revealed their unique ability to be reprogrammed into progenitor cells, which, in turn, can differentiate into neuronal and photoreceptor-like cells. This process could occur without the growth factors involved, although the presence of FGF2 and IGF2 in the medium stimulated cell differentiation in the neural direction [145].

The process of the natural reprogramming of iris pigmented epithelial cells into retinal cells has not been recorded from the adult vertebrate eye in vivo. Only once we could observe the formation of retinal-like stratified structures from the cells of the ciliary region of the inner iris layer (iris pars ciliaris) in an experiment with the implantation of an RPE fragment from the eye of adult newt into the eye cavity of the same animal [146]. However, this phenomenon, as well as the ability of the RPE to be reprogrammed and fully regenerate the stratified retina in vivo, is probably characteristic of only these animals, which are known to have the overall highest regenerative capabilities.

## 5. Müller Glia

In addition to the vertebrate RPE, CB, and iris cells discussed above, the currently widely studied Müller glia (MG) are also a latent differentiated RRCS [7,147,148,149]. Observations of neural stem cells found in the vertebrate brain, with a phenotype close to the glial one, have become an incentive to study MG as an RRCS [150]. As a result, the MG cell population is currently considered as having the greatest potential for retinal repair [145,151,152]. The findings below have heightened our hopes that MG cells could help to overcome the severely stretched regenerative capacity of higher vertebrate retina.

MG cells are arranged radially in the inner retina (Figure 1 and Figure 2). Their bodies are located in the inner nuclear layer, i.e., the layer of interneurons, and the long processes extend to the outer and inner limiting membranes of the retina (Figure 2). MG are highly specialized cell populations that perform a wide range of functions, including structural and neurotrophic ones, are integrated in the system of retina cleaning and light perception, and have connections with all types of retinal neurons [153]. Furthermore, an extensive amount of experimental data obtained using various animal models provide evidence that MG can behave as a population of neural progenitor cells [154] (Figure 8).

The ability of the differentiated MG cells of the mature retina to proliferate in vitro, as well as in pathological conditions in vivo, has been proven [155,156]. In higher vertebrates, MG cells usually respond to damage by reactive gliosis, which is manifested as proliferation and an increase in the MG population and in the thickness of the long processes of these cells. Such behavior of MG cells accompanies many pathological states of the retina [157]. It is also known that several TFs characteristic of progenitor cells of the retina and immature MG (six3, pax6, rx1, olig2, and vsx2) are re-expressed in mature MG cells after retinal damage, thus demonstrating the process of reduction in their differentiation level.

In the case of the surgical excision of the zebrafish retina, MG cells in regions not affected by the damage proliferated and produced retinal progenitors that differentiated into photoreceptors: cones and interneurons [158]. As has been shown in experiments on fish with thermal or light-induced lesions of the retina, causing a loss of photoreceptors, MG cells re-enter the cell cycle at 1 h post injury, as evidenced by PCNA expression, with the subsequent up-regulation of the expression of stem and progenitor cell proteins [2,159,160]. Even in the healthy retina, MG cells are known to express genes, in particular Pax6, Dkk3, and Chx1, whose action is often associated with retinal progenitor cells [161,162,163]. 

In fish, whose eyes grow throughout life, MG cells maintain the lineage of photoreceptors (rods) and can provide a progenitor population for photoreceptors and ganglion cells during the regeneration [7]. In post-hatch birds, the regenerative responses of these cells—proliferation and the expression of “developmental” TFs—can also be observed, which, however, requires the impact of signaling molecules such as IGF, FGF2, and RA [164,165,166].

The MG cells in postnatal mammals and the dividing retinal progenitors during development exhibit a high degree of transcript overlapping, which is much more significant than that of other retinal cells differentiated in postnatal development. Thus, after the first postnatal week in mice, 68% of the genes identified as specifically expressed in MG cells were found to be enriched in mitotic progenitor cells based on their in situ hybridization pattern, while photoreceptor-specific genes accounted for only 14% [161]. It is assumed that late retinal progenitors can differentiate into Müller glial cells without entering the irreversible state that occurs when retinal neurons mature. 

In vivo differentiated MG cells of the mouse retina are characterized by the expression of genes responsible for division [167]. Under the effect of exogenously added components of the EGF, FGF, Notch, Wnt, and SHH signaling pathways, MG retinal cells in higher vertebrates can be induced to re-enter the S-phase and exhibit certain properties of retinal progenitors [160,167,168,169,170,171] In a rat model of primary photoreceptor degeneration, the production of retinal progenitors occurred after the transplantation of in vitro-obtained human MG with stem cell characteristics into the retina. These progenitors were capable not only of migration, but also of subsequent integration and functioning in the outer nuclear layer of the recipient’s retina [172].

Data obtained on the human retina indicate the expression of many genes of retinal progenitors, including *Pax6* and *Sox2*, in the retinal cells in response to retinal damage and the stimulation of proliferation [173]. In the normal adult human retina in vivo, MG cells show the expression of the proteins CD117 and CD44, which are markers associated with stem cells, along with vimentin, which is characteristic of glial differentiation [174].

The “developmental” profile of genetic expression recorded from various models, as well as the ability to proliferate, eventually constitute the basis of MG reprogramming into proliferating progenitors and, subsequently, retinal neurons in permissive environmental conditions. It is worth noting that, in addition to the above facts, the gene expression profile in differentiated MG cells, including such genes as *ApoE*, *Glul*, *Kcnj10*, and, *GFAP* reflects their specialization, which is sufficient to perform the functions necessary for the retina. The expression of the components of the cell cycle machinery and the Notch signaling pathway, as well as growth factors, chemokines, and lipoproteins, all provide functions performed by MG cells, including communication with retinal neurons, and up to the modulation of their neuronal activity [162]. 

Thus, when considering the MG cells as an RRCS, we observe a certain combination of genes operating in the gene expression profile, associated with both retinal progenitor cells and the functional specialization of MG proper. If so, what then could be the mechanism that blocks the implementation of the further stages of the MG cell reprogramming pathway, the segment of conversion from a progenitor cell to a differentiated neuron? The epigenetic features of these cells are candidates for the role of such a regulator, along with both the already known regulatory factors of the MG environment and the currently detected ones, such as the factors of post-transcriptional regulation by miRNA [175]. 

Despite the fact that the role of epigenetic changes associated with reprogramming and other MG regenerative responses has not yet been sufficiently understood, the current research is gradually developing in this direction to address the issues above. Data have been obtained on the epigenetic changes in fish retinal MG cells during their conversion to poorly differentiated neuronal progenitors, in particular, the changes in the DNA methylation landscape [25]. The study was based on the well-known fact that the expression of genes encoding the pluripotency TFs (Oct4, Klf4, Sox2, c-Myc, Lin28, and Nanog) correlates with “open”, accessible chromatin, while the repression of pluripotency genes against the background of less accessible, condensed heterochromatin, vice versa, occurs in somatic cells [176]. The reprogramming of the zebrafish retinal MG cells, however, showed that this pattern is not reproduced in the MG conversion model. In the initial specialized cells of the fish retinal MG and during the first days of cell conversion, the hypomethylation (accessibility) of chromatin was observed; then the level of methylation increased. The conclusion was made that during the reprogramming of MG cells into progenitors, the genome actually undergoes dynamic changes in the level of DNA methylation, whereas the hypomethylation of DNA, observed initially and at the beginning of MG conversion, may indicate the initial “readiness” of these cells to develop in a different, neural direction [25]. All the observations above agree well with the assumption that MG cells maintain the plasticity of differentiation that allows them to be reprogrammed and replace damaged, dying retinal cells by involving additional epigenetic modifications in the process [25,177]. 

By creating adequate epigenetic conditions and providing the overexpression of the *Ascl1* gene in the neurotoxin-damaged retina of adult mice, it became possible to induce neurogenesis in the MG population at the stage of the animals’ development when the neurogenic potential had been exhausted [178]. It was found that the relationship of the neurogenesis carried out by MG cells with the age of mice is associated with the loss of access to chromatin by the specially transfected *Ascl1* gene, necessary for the activation of MG cells. This was due to blockage by some epigenetic factor determined by the animal’s age. The use of the highly sensitive method of integrative epigenomic analysis, that identifies closed and open regions of chromatin, has shown that the histone deacetylase inhibitor can increase access to key regions of MG cell genes and thereby contribute to the production of MG progeny cells, which are progenitors for retinal regeneration. These progenitors demonstrated the expression of marker proteins of interneurons, and also formed synapses with pre-existing retinal neurons of the retina [177]. 

Recently, Dvoriantchikova and co-authors [178] conducted a study that helps understand the level of epigenetic plasticity of MG cells that allows them to be reprogrammed and differentiate into retinal neurons under permissive conditions. In transcriptome studies, it has been found that only a small group of genes encoding proneural marker proteins and the genes whose expression is associated with the cell cycle undergo significant changes in expression during development, in particular, in the process of the differentiation of embryonic progenitors into the adult MG. Furthermore, the expression of the genes associated with immune response increases in the process of MG cell differentiation. Based on the actual invariability of the expression of other genes, the authors have noted the similarity between mature MG cells and late-born progenitor cells in the mouse retina, thus confirming the previously made observations [161,162,163]. Data describing the epigenetic profile of MG cells indicate that it is epigenetically close to the phenotype of retinal progenitors, including precursors of late-born neurons that are bipolar cells and rod photoreceptors. The authors [178] suggest that MG cells, for this reason, should have no epigenetic barriers preventing their reprogramming and differentiation into late-born retinal neurons. Their ChIP-seq data also show that the obstacles for the regeneration of the mammalian retina from MG after injury may be due to the repressive chromatin state of many genes required for the late stages of reprogramming, i.e., the stages of maturing neurons’ specialization. It is assumed that the epigenetic plasticity of adult mammalian MG can be restored at this stage in the presence of some transcription factors and DNA demethylase activity. This, in turn, should allow MG cells to be involved in cell replacement, that is, the regeneration of the retina after damage in mammals [179]. These and other recent works in this direction [163,179,180,181] give some hope of resolving the issue of retinal regeneration from MG as an RRCS.

Thus, the data obtained allow us to characterize MG cells as an RRCS, which are ready, not only to initiate proliferation in conditions of retinal pathology, but also to be reprogrammed into neurons to replace lost cells for retinal regeneration. This “readiness” is manifested both as the expression of a number of genes associated with retinal progenitors and as the epigenetic landscape resembling that of progenitor cells. We should emphasize here that MG cells are specialized with a full set of genes whose expression is required to perform a wide range of MG functions. It should also be noted that the possibility of the conversion of the MG cells into neural progenitor cells and neurons is predetermined by the common origin of neurons and glia in development. It is known that a common precursor for neurons and MG is long present in the developing mouse retina [182] and that MG is one of the last, late-maturing cell types in the retina [183]. Recently, Bachmann et al. [184] showed that, during the in vitro formation of the avian retina, there exists a precursor producing both cholinergic amacrine cells and Müller cells, developing in close association. Together, they are involved in the initiation of inner retinal layer (IPL) network formation [184]. Therefore, in the central retina, similarly to the ciliary region, the late maturation of some cell populations, against the background of retinal neurons that have already reached specialization, can be one of factors responsible for the retention of a number of progenitor properties by RRCSs.

## 6. Conclusions

A number of retinal degenerative diseases result in the death of retinal neurons, leading to reduced vision and, in extreme cases, blindness. To date, some endogenous cell populations have been identified as potential sources for retinal regeneration (RRCSs), which, due to their autologous origin, have substantial advantages over foreign cells transplanted for retinal repair. The category of RRCSs includes cells of the eye periphery: the ciliary marginal zone (CMZ) and the ciliary body (CB), which is topologically similar to the CMZ, the retinal pigment epithelium (RPE), the iris, and Müller glial (MG) cells.

In lower vertebrates, the above-listed cell populations are involved in the growth and regeneration of the retina and demonstrate the ability to proliferate, differentiate (CMZ), or to re-enter the cell cycle and be reprogrammed in the neural direction (the RPE, iris, and MG) in vivo. In higher vertebrates, e.g., mammals and humans, such an ability of these cell populations is lost, but it can be observed in conditions permissive for these processes in vitro. A wide range of extrinsic factors has been found which provide and control the conversion of RRCSs in adult higher vertebrates in vitro and in vivo. This has become possible due to the accumulated knowledge about the molecular mechanisms of eye formation and an understanding of the phenotypic status of cell resources for retinal regeneration determined by the spatio-temporal features of their development.

The analysis of the molecular and genetic properties of RRCSs indicates that RPE, CB, iris, and MG cells in adult higher vertebrates in the course of conversion, specifically provoked by in vitro conditions or, in some cases, by retinal damage in vivo, are capable of acquiring a specific phenotype with features of progenitor one (a specific progenitor cell-like state). In this case, the complete loss of differentiation features is not observed, and the cells often show only some traits of progenitor cells, with the simultaneous retention of features of the original differentiation for a certain period of time. This is usually manifested as the initiation of the expression of some genes characteristic of eye and retinal progenitors in development, as well as genes and factors that control entry into the cell cycle.

The data concerning the epigenetic regulation of the conversion process, known to date, indicate that RRCSs have an epigenetic landscape that is in a certain state of readiness (flexible state) for the differentiation of cells in the retinal direction and resembling that of retinal progenitor cells in development. However, the stable high specialization of RRCSs and also the age of mammals, leading to changes in this epigenetic landscape, impose a ban on the full manifestation of the potential and regenerative responses of RRCSs. Thus, a deeper knowledge of RRCS biology, in terms of both the molecular genetic profile and epigenetic landscape, along with an understanding of the cellular and system environment factors that regulate these processes, can help resolve the issue of the regeneration of the retina from its endogenous cell resource.

## Figures and Tables

**Figure 1 biomedicines-08-00208-f001:**
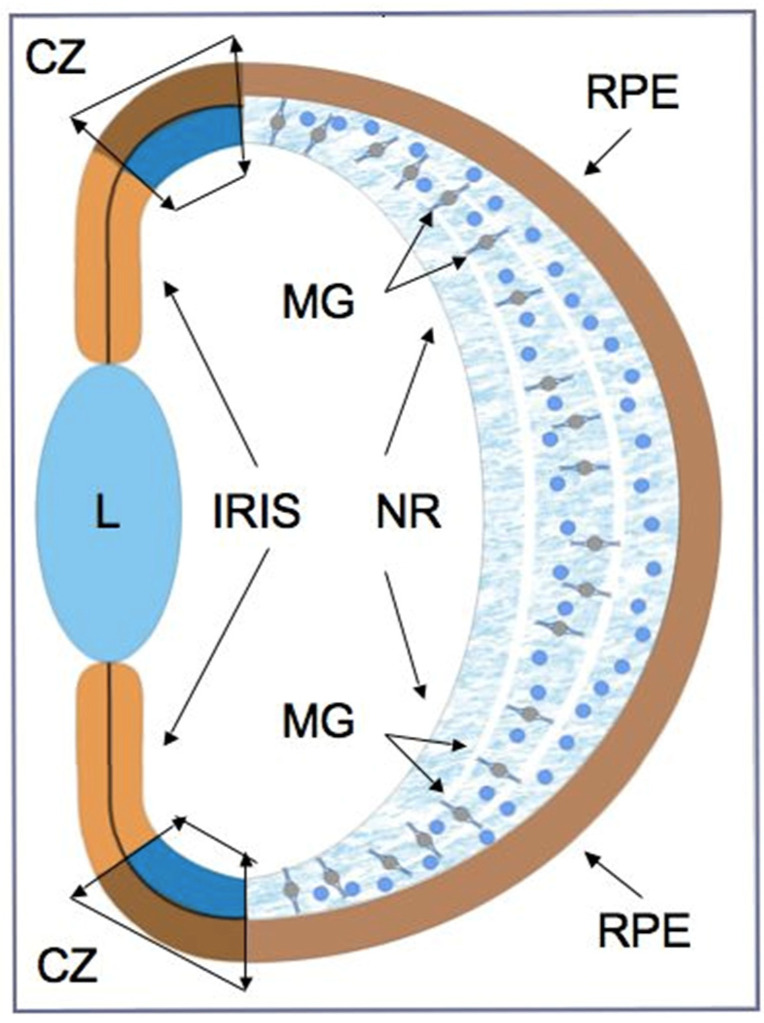
Schematic diagram of retinal regeneration cell sources (RRCS)s’ locations in the eye of adult vertebrates (summarized data). RPE—retinal pigment epithelium cells (latent differentiated); NR—neural retina; MG—Müller glial cells (latent differentiated); Iris cells (latent differentiated); CZ—eye ciliary zone: ciliary marginal zone (CMZ) in low vertebrates (stem cells and low differentiated precursors) and ciliary body (difined CB cells in mammals (latent differentiated); L—lens.

**Figure 2 biomedicines-08-00208-f002:**
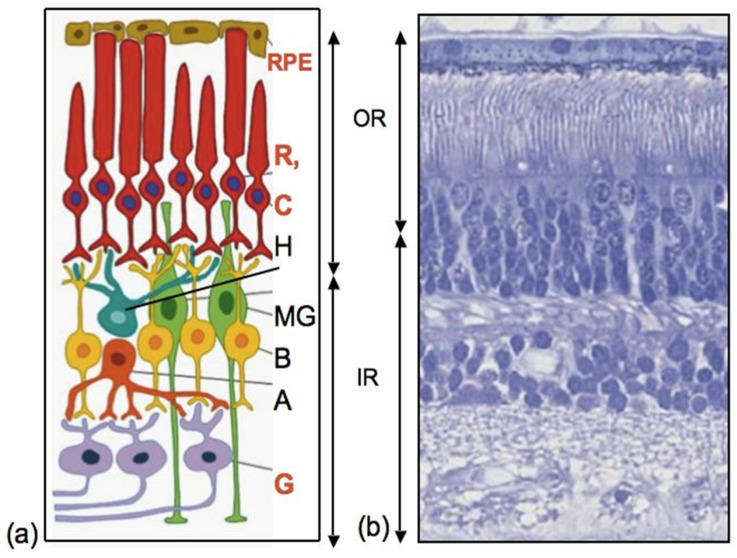
The structure of the retina of vertebrates. (**a**) Schematic diagram, (**b**) histological picture of mouse eye retina. (OR): RPE—retinal pigment epithelium; R—rods, C—cones; H—horizontal cells; (IR): MG—Müller glia; B—bipolars; A—amacrine cells; G—ganglion cells. The most common damage target cells are highlighted in red. OR—outer retina, IR—inner retina.

**Figure 3 biomedicines-08-00208-f003:**
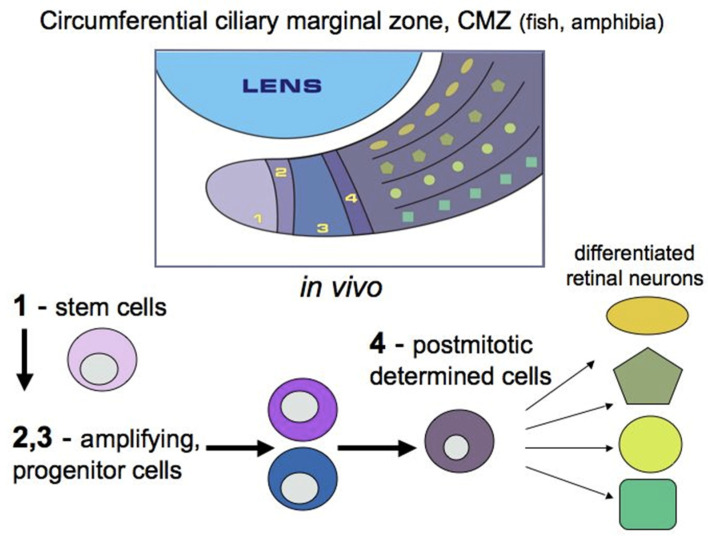
Postembryonic growth of the eye through cell division at the ciliary marginal zone in lower vertebrates. 1–4: The CMZ can be divided into several zones, from peripheral to central, which reflect different stages of the development of retinal stem cells. (In accordance with [31]).

**Figure 4 biomedicines-08-00208-f004:**
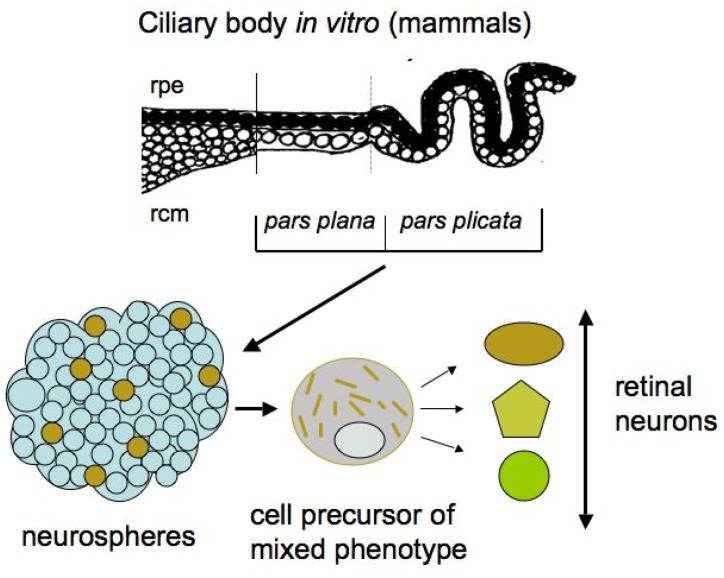
Potencies of mammalian ciliary body cells that produce neurospheres containing cell precursors of mixed phenotypes capable of differentiating into retinal neurons. rpe—retinal pigment epithelium; rcm—retinal ciliary margin. More details are in the text.

**Figure 5 biomedicines-08-00208-f005:**
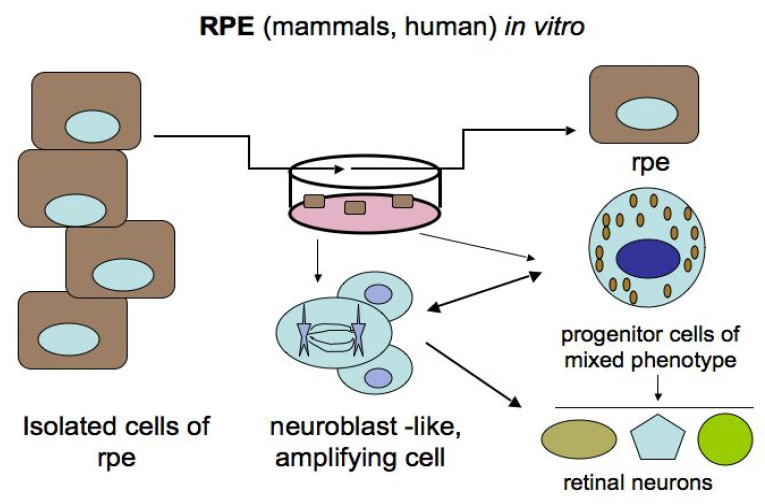
Phenotypic plasticity of mammalian retinal pigment epithelium cells in vitro. Different levels and types of differentiation were observed, dependent on cell culture conditions. More details are in the text.

**Figure 6 biomedicines-08-00208-f006:**
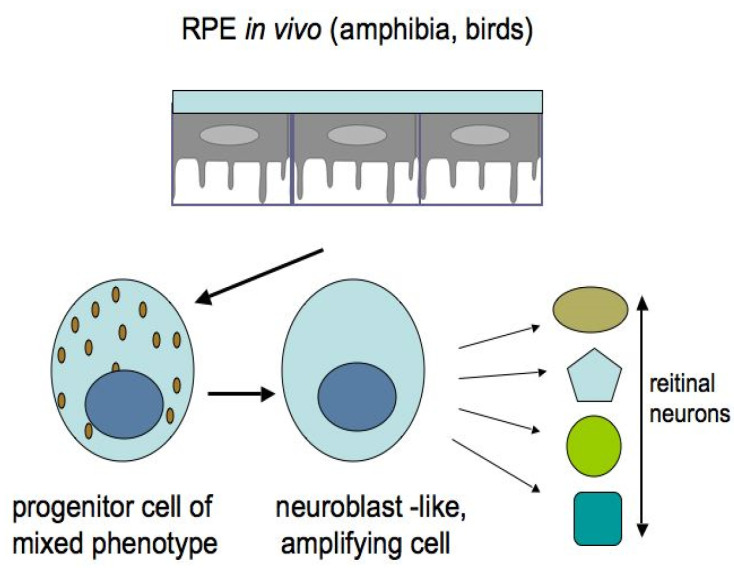
Retinal pigment epithelium cells of adult amphibia and bird embryos are capable of natural reprogramming in vivo. The picture shows the sequence of the main steps in the formation of retinal neurons.

**Figure 7 biomedicines-08-00208-f007:**
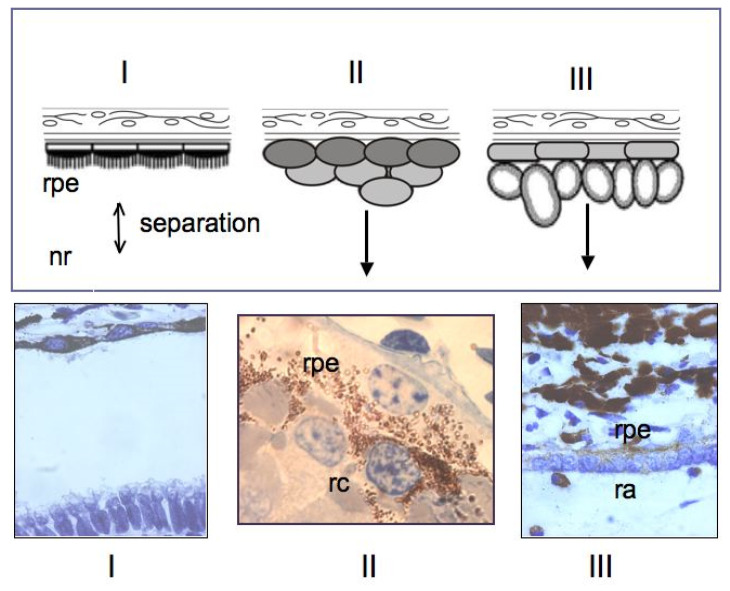
Early stages of newt retinal pigment epithelium (rpe) reprogramming in vivo. I—neural retina is separated from the RPE; II—some RPE cells come out of the layer and form retinal anlage (III). Others divide and return to their original differentiation. rc—reprogramming cells, ra—retinal anlage.

**Figure 8 biomedicines-08-00208-f008:**
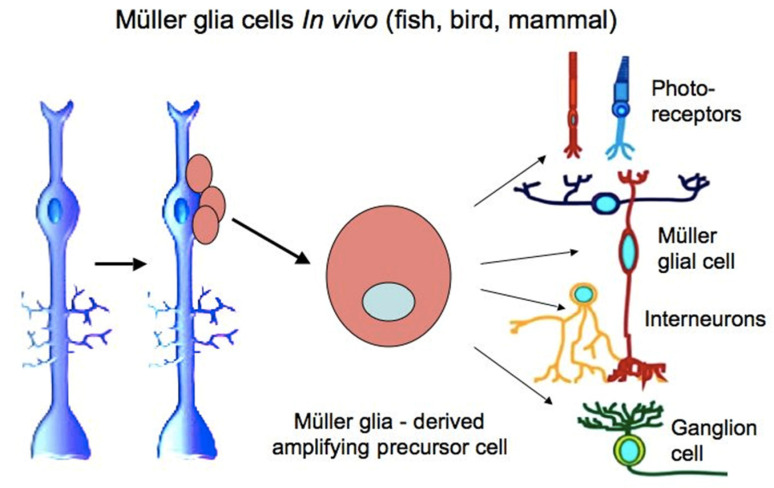
Schematic diagram showing the main steps towards the production of retinal neurons by Müller glial cells in vivo. Data obtained on various animal models are summarized.

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
