# Peer review of "Potential Endogenous Cell Sources for Retinal Regeneration in Vertebrates and Humans: Progenitor Traits and Specialization"

_biomedicines, 2020, doi:10.3390/biomedicines8070208_

Round 1

Reviewer 1 Report

Grigoryan Review: Overall, this is an excellent review article on the issue of retinal tissue regeneration in lower and higher vertebrates by investigating possible retinal regeneration cell sources (RRCS). Having worked on retina regeneration over most of her entire career, Dr. Grigoryan has a vast overview on the relevant experimental and medical literature, certainly a fundus from which this review profits profoundly. Hence, this article is most instructive and timely for a wide audience of researchers in the ophthalmic sciences. In particular this is so for biomedical research into finding regenerative therapies for blinding diseases.

After a general introduction (incl. eye diseases, structure of retina, signaling pathways and relevant transcription factors, TFs), the extensive bulk of reviewed material is divided into section 2. Eye ciliary zone (incl. sub-sections on CMZ and ciliary body), 3. Retinal pigment epithelium (RPE), 4. Iris, 5. Müller glia, and 6. Conclusion. The author compares regenerative capacities of these major eye constituents in various model species (fish, amphibia, birds, mammals incl. human) on cellular and histologic levels, and attempts to explain their specifics by extensive molecular literature findings, e.g. signaling pathways, transcription factors and genetic profiling. I found the up-dated discussion of possible mechanisms of epigenetic plasticity particularly informative and relevant (l. 575ff). Taken together, this review conveys a tremendous amount of valuable information. I could not see many gaps in her long list of 178 citations (except my comment below). The article has a clear structure and is well written, which makes reading easy. The 8 figures (mostly schematics) are all appropriate and easy to comprehend (but see remark below); typos are very few (see below).

My comments and Errors:

  1. Early literature on regeneration of retina remains selective. In particular, seminal work on reaggregates from avian eyes (retinospheroids; nowadays called organoids) were most revealing both in respect to RPE and on Müller cell (MCs) effects on regeneration of retinal tissue in vitro. A continuing series of studies beginning in the early 80ies showed – historically for the first time (Vollmer et al., 1984; reviewed in Layer & Willbold in Int Rev Cytol 1993, Prog Ret Eye Res 1994; Layer et al. Neuroreport 2001; TINS 2002) - that i) RPE is decisive to organize a correctly and fully laminar retinal tissue that ii) RPE from central vs. peripheral eye acts differently (Layer & Willbold, 1989), and that iii) RPE cells can be a source of in vitro production of retinal neurons. Further studies using this in vitro spheroid approach revealed that a) MCs also can function as a source of stem cells, b) have a strong lamina organizing effect (Willbold et al., 2000) and c) are involved in the earliest network formation of an IPL in vitro (Bachmann et al., 2020, IOVS). We feel that citing this seminal in vitro work would complement this excellent review article in several respects.
  2. 1 is presented as mirror-image: correct;
  3. 2b, legend: mention animal of histological picture;
  4. Arrows indicating “IR” vs. “OR” do not show the same retinal fractions in a and b. I personally would prefer those from b, since HZ belong to the outer retina;
  5. Lines 217-220 are misleading: I guess it should read …have no retinal region whose cells are capable…??
  6. Sentence following 327 is confusing: split and clarify;
  7. Distinction between circumferential (Fig. 3) and ciliary marginal zone should be defined more clearly, if there is any;
  8. 604ff, precursor for neurons and glia: cf. Bachmann et al. (IOVS, 2020) showing that during in vitro formation of avian retina, there exists a precursor producing cholinergic ACs and MCs, and that these two cell types together are responsible for earliest organization of the inner retinal network (IPL);
  9. 626ff: the last paragraph of Conclusion on epigenetic effects appears too long (too detailed) at the end of this review article.

Typos:

  • 100: type…Circumferential …
  • 148: type…Xenopus…
  • 645: type …resources.
  • In Fig. 8, label as plural: fish, birds, mammals

Author Response

First of all, let me express my gratitude for the careful analysis of the review and very useful comments made.

Below: answers to the questions and corrections made according to your recommendations.

1). Early literature on regeneration of retina remains selective. In particular, seminal work on reaggregates from avian eyes (retinospheroids; nowadays called organoids) were most revealing both in respect to RPE and on Müller cell (MCs) effects on regeneration of retinal tissue in vitro. A continuing series of studies beginning in the early 80ies showed – historically for the first time (Vollmer et al., 1984; reviewed in Layer & Willbold in Int Rev Cytol 1993, Prog Ret Eye Res 1994; Layer et al. Neuroreport 2001; TINS 2002) - that i) RPE is decisive to organize a correctly and fully laminar retinal tissue that ii) RPE from central vs. peripheral eye acts differently (Layer & Willbold, 1989), and that iii) RPE cells can be a source of in vitro production of retinal neurons.

I have included this information in the RPE (lines 365-373, italic) section.

I whould like also to provide a link to our another article, where the works of this laboratory are also cited: Novikova, Y.P., Poplinskaya, V.A. & Grigoryan, E.N. Organotypic Culturing as a Way to Study Recovery Opportunities of the Eye Retina in Vertebrates and Humans. Russ J Dev Biol 51, 31–44 (2020). https://doi.org/10.1134/S1062360420010063

2). Figure 1 is presented as mirror-image: correct; figure 1 corrected.

3). 2b, legend: mention animal of histological picture; legend to figure 2 rewritten.

4). Arrows indicating “IR” vs. “OR” do not show the same retinal fractions in (a) and (b). I personally would prefer those from b, since HZ belongs to the outer retina; I tried to made OS and IR in (a) and (b) equal. It was not easy, the only thing I could do it’s to replace the letter (H) marked Horizontal cells to OR fraction (arrows) in the picture and legend.

5). Lines 217-220 are misleading: I guess it should read …have no retinal region whose cells are capable…?? Sentence rewritten (lines 253-256).

6). Sentence following 327 is confusing: split and clarify; Sentence rewritten (lines 356-359).

7). Distinction between circumferential (Fig. 3) and ciliary marginal zone should be defined more clearly, if there is any. For that I included the word “Circumferential” to the title of the picture 3 and “ciliary” to the heading 2.1., since there is no contradiction between these two words. “Circumferential” and “ciliary” are used equally in literature and complement each other sometimes.

8). 604ff, precursor for neurons and glia: cf. Bachmann et al. (IOVS, 2020) showing that during in vitro formation of avian retina, there exists a precursor producing cholinergic ACs and MCs, and that these two cell types together are responsible for earliest organization of the inner retinal network (IPL); This information (thanks for it!) included in the text of MG section (lines 667-669, italic).

9). 626ff: the last paragraph of Conclusion on epigenetic effects appears too long (too detailed) at the end of this review article. The part of last paragraph in Conclusion was removed.

10). In Fig. 8, label as plural: fish, birds, mammals. Chaging the title in Fig.8, I decided to use: «fish, bird, mammal» as names of animal classes. Corrected.

11). Typos – corrected and marked as comments to the paper

Thanks for your comments, with respect, E.N.Grigoryan

Reviewer 2 Report

This manuscript by Grigoryan reviewed different kinds of retinal regeneration cell sources (RRCSs) in terms of their function, transcriptional and epigenetic characteristics, and regeneration properties. The author should discuss the following questions:

  1. What is the definition of RRCSs? What are the criteria that a certain type of cell could be counted as a RRCS?
  2. The author should briefly discuss all the current methods to treat retinal disorders? And What are the advantages of RRCSs compared to other method such as induced pluripotent stem cells?
  3. In the introduction, the author should discuss how RRCSs potentially could help treat retinal diseases? What is the state of art for this?
  4. The author discussed different cells in this review, it would be better if the author could discuss the specific role of each RRCS in treating retinal disease?
  5. The author should make a table to summarize and compare the RRCSs, in terms of their morphology, function, transcription and epigenetic markers, regeneration property, and signaling pathways to promote and block their regeneration.
  6. In the second half of section 2.1, the author discussed CMZ in mammals and humans; in section 2.2, the authors discussed CB. What are differences between “CMZ in mammals and humans” and CB?

Author Response

First and foremost I am very thankful to the reviewer for attentive reading of the paper and valuable comments made. Below: my answers and amendments to the text. The latter are marked in the margin of the paper as a comments.

1). What is the definition of RRCSs? What are the criteria that a certain type of cell could be counted as a RRCS?

Following paragraph is added to the Introduction:

“Several criteria can be set out to attribute one or another eye cell population to the category of RRCSs. First, it is the previously confirmed their involvement in retinal regeneration after damage or under experimental conditions. Second, it is the experimentally revealed facts indicating the eye cells’ potency to re-express “developmental” genes and organize a permissive epigenetic landscape for this, i.e., the conditions underlying and determining the initiation and progress of regenerative responses. The RRCSs criteria at the cellular level may include the ability to proliferate, migrate, and be integrated in the pre-existing retinal structure. An additional criterion is also the history of maturation of these cell populations in development that a priori suggests the possibility of their reprogramming into certain cell types of retinal neurons. This does not mean that the criteria above are fully applicable to all cells considered as candidates to RRCSs. Their set depends on the species of animal, age, conditions of eye pathology or experiment, and, finally, the degree of knowledge of the issue on a particular animal model". (lines 52-62).

2). The author should briefly discuss all the current methods to treat retinal disorders? And what are the advantages of RRCSs compared to other method such as induced pluripotent stem cells?

The corresponding paragraph inserted in the Introduction: lines 31-41).

«As regards the existing methods of treatment of degenerative retinal diseases associated most frequently with death of photoreceptors (age-related macular degeneration (AMD), retinitis pigmentosa (RP), proliferative vitreoretinopathy (PVR)) and ganglion cells (glaucoma, optic nerve thrombosis) at their late stage, they have not been proven to date despite considerable effort and significant therapeutic advances. The currently known methods include drug therapy, liposomal and nanotherapy, neuroprotection, surgery, immune therapy, gene therapy, and cell transplantation, including IPSC technology and 3D retinal organoid production, etc. The study of RRCSs, without rejecting any of the existing treatment methods, provides clues to the development and application of new approaches to retinal regeneration based on the recruitment of endogenous cell sources located in eye tissues of animals and humans.

3). In the Introduction, the author should discuss how RRCSs potentially could help treat retinal diseases? What is the state of art for this?

My answer (below) added to the Introduction. Lines - 121-126.

‘In the review, I made an attempt to show that RRCSs exhibit the genetic and epigenetic potencies: proliferation, reprogramming, and subsequent differentiation towards the phenotypes of affected or dead cells for their replacement. The key to implementation of these potencies and identification of ways of RRCS-based treatment in case of disease is the complete knowledge of both the biology of these cells and the factors that regulate cell behavior or, in other words, the ways of induction and targeted regulation of regenerative responses”.

 4). The author discussed different cells in this review, it would be better if the author could discuss the specific role of each RRCS in treating retinal disease?

This is mentioned in each section in some or other way: for CB – lines 289 -293, RPE – lines 371-373; 491-494, MG – lines 529-531; 653-656. However, taking into account that these studies on RRCSs biology are currently in progress and are aimed at possible application in future, the statement of the specific role of each RRCS in treating retinal diseases is still premature. Nevertheless, in the section on MG, I emphasize the most important role and a greater, as compared to other RRCSs, potential of using MG to treat retinal diseases. Lines -529-531.

 5). In the second half of section 2.1, the author discussed CMZ in mammals and humans; in section 2.2 the authors discussed CB. What are differences between “CMZ in mammals and humans” and CB?

In the review, I indicate that adult mammals and humans do not have an eye zone (CMZ) similar to that in fish and amphibians. Also, I mention that CB of higher vertebrates and humans is exclusively topological analog of CMZ. Those data that indicate “traces” of CMZ in mammals (lines 207-213) only refer to a period of eye development. There is also a mention of the region of pars plana (orbicularis ciliaris), the non-pigmented single-rowed epithelium, which is topologically a continuation of the retina (lines 214-225). This is a very small cell population bordering on CB. It may be reffered either to as marginal region of neural retina or CB. Pars plana can be regarded as CMZ (in it accepted understanding) very conditionally only. That is added to clarify (lines 218-219).

6). The author should make a table to summarize and compare the RRCSs, in terms of their morphology, function, transcription and epigenetic markers, regeneration property, and signaling pathways to promote and block their regeneration.
This is a very good recommendation, thank you. However, compiling such table is a subject of separate, very large comparative and analytical work that can require substantial effort to include all the above aspects of biology of each RRCS type. Information on the recommended sections of a summary table is represented in the literature very broadly, which makes it extremely difficult to provide its adequate overview in a single table.
Thanks again for your work and suggestions, with respect, E.N.Grigoryan

Round 2

Reviewer 2 Report

I don't have any more comments for this manuscript. Congratulations to a great paper!